# OpenReview forum: "SENTINEL: A Multi-Level Formal Framework for Safety Evaluation of LLM-based Embodied Agents"
_ICLR.cc/2026/Conference — Submitted to ICLR 2026_

### Official Review · Reviewer_eNbM · 2025-10-26

**Soundness:** 2
**Presentation:** 3
**Contribution:** 1
**Rating:** 2
**Confidence:** 5

**Summary:**

This paper introduces SENTINEL, a framework for evaluating the physical safety of LLM-based embodied agents. The authors propose a multi-level verification pipeline that operates at three levels of abstraction:Semantic-level: Evaluates an LLM's ability to translate natural language (NL) safety requirements into formal Linear Temporal Logic (LTL) formulas (i.e., $NL \rightarrow LTL$).Plan-level: Verifies if the agent's high-level action plan (a single sequence) satisfies the LTL constraints.Trajectory-level: Samples multiple execution trajectories from the agent, merges them into a computation tree, and uses Computation Tree Logic (CTL) (e.g., checking $AG(\phi)$) to verify that all possible execution paths are safe.The primary contribution is the proposal of this formal, multi-level approach, which aims to provide a more rigorous safety evaluation than existing methods that rely on heuristic rules or subjective LLM-based judgments.

**Strengths:**

The paper addresses the physical safety of LLM-based embodied agents. This is a critical, high-impact research area and a significant bottleneck for the trustworthy, real-world deployment of these systems.

**Weaknesses:**

**1. Questionable Motivation and Limited Novelty**

* **Unnecessary Complexity:** The framework's core premise of translating natural language (NL) rules into formal temporal logic (TL) adds a complex layer that may be unnecessary. This translation step itself is a significant source of failure, as shown by the paper's own semantic evaluation, and complicates the process for LLMs that are designed to reason over NL directly.
* **Insufficient Justification:** The paper fails to justify *why* this translation is necessary. It does not explain why an LLM capable of understanding complex NL task instructions cannot also be evaluated directly against NL safety constraints.
* **Unmet Burden of Proof:** The authors do not demonstrate that their complex, TL-based evaluation is more effective at detecting safety violations than a simpler, direct NL-based evaluation. The assumption that TL is superior to NL for this task remains unproven.

**2. Doubts Regarding the Methodology**

* **Decoupled Hierarchy:** The "multi-level" framework does not match an agent's runtime decision flow (Instruction $\rightarrow$ Plan $\rightarrow$ Action). "Level 1" (NL-to-TL translation) is merely an offline pre-processing step, completely decoupled from the agent's reasoning. A true hierarchical evaluation should first assess the agent's ability to identify risks from the initial instruction *before* planning.
* **Redundant Logic:** The switch from LTL to CTL is poorly justified. Verifying that all *n* sampled trajectories are safe can be achieved by *n* independent LTL checks. Merging paths into a tree for a single, more complex CTL check is logically equivalent, and its added benefit over the simpler approach is not explained.
* **Offline-Only Assessment:** The framework is a purely *post-hoc* auditing tool, not a real-time safety mechanism. It evaluates trajectories *after* they have been fully sampled, offering no capability to intervene or prevent an unsafe action as it happens, which severely limits its practical utility.

**3. Lack of Evaluation on Mainstream VLA Models**

* **Outdated Model Scope:** The evaluation is restricted to LLMs, ignoring the more relevant Vision-Language-Action (VLA) models that dominate modern robotics.  VLA models, which fuse vision into their decisions and directly output control params, will have different failure modes than text-only llms. By excluding VLAs, the paper fails to assess the framework's applicability to state-of-the-art agents or to compare safety performance across architectures.

**4. Absence of Real-World Robotic Evaluation**

* **Simulation-Only Validation:** The experiments are confined to simulation, which is insufficient for evaluating *physical* safety. This approach ignores real-world physics, sensor noise, and actuator latencies that are critical to safety.
* **Ignoring the Sim-to-Real Gap:** The paper makes no attempt to bridge the "sim-to-real gap." A plan verified as "safe" in simulation could fail catastrophically in the real world, and without validation on physical hardware, the claims about "physical safety" are not credible.

**5. Missing Critical Experimental Details**

* **Undefined Data Curation:** The paper fails to describe the *criteria* or *methodology* used to extract the "safety-related subset" of tasks from VirtualHome and ALFRED. This makes it impossible to judge if the dataset is representative or biased.
* **Unspecified Dataset Size:** The total size (N=?) of the task datasets is never reported. Without this, the reported percentage-based metrics are statistically meaningless and their robustness cannot be assessed.
* **Ambiguous Notation:** Key formal notation is left ambiguous or undefined. For instance, the definition of "C" is not clearly defined in line 140-141, creating confusion in a paper that should be formally precise.

**Questions:**

See above.

---

> ### Author Response · Authors · 2025-11-22
> **Response to Reviewer eNbM Part 1/4**
>
> Thank you for reviewing our paper and providing comments. In the following responses, we clarify several misunderstandings and address the questions.
>
> > ## Questionable Motivation and Limited Novelty:
> > #### Unnecessary Complexity: The framework's core premise of translating natural language (NL) rules into formal temporal logic (TL) adds a complex layer that may be unnecessary. This translation step itself is a significant source of failure, as shown by the paper's own semantic evaluation, and complicates the process for LLMs that are designed to reason over NL directly.
> > #### Insufficient Justification: The paper fails to justify why this translation is necessary. It does not explain why an LLM capable of understanding complex NL task instructions cannot also be evaluated directly against NL safety constraints.
> > #### Unmet Burden of Proof: The authors do not demonstrate that their complex, TL-based evaluation is more effective at detecting safety violations than a simpler, direct NL-based evaluation. The assumption that TL is superior to NL for this task remains unproven.
> >>
> >> Below, we first clarify that the use of formal temporal logic (TL) is a fundamental  requirement for rigorous safety evaluation. We then explain the role of semantic evaluation in measuring an LLM agent's capability to correctly interpret safety rules.
> >>
> >>**Motivation for grounding safety evaluation in formal temporal logic:** Natural language (NL) is inherently ambiguous and incomputable. It often contains hidden context and is difficult to verify algorithmically. Consequently, relying on NL constraints leads benchmarks to use ``LLM-as-a-judge'', which lacks rigorous safety definitions and evaluations and is limited in their trustworthiness when assessing agent safety, as discussed in **Section 1 (Introduction)** of the paper. This is in fact a key motivation for our framework to define ground-truth safety constraints in TL and develop formal procedures to evaluate them. This mirrors why programming languages use formal syntax and semantics rather than natural language: formal semantics provide the precise, unambiguous specifications required to develop executable programs and algorithmically verify their correctness. More specifically in SENTINEL, we chose TL because it provides precise and diverse formal semantics for invariants, ordering, and timing properties, enabling rigorous qualification and quantification of system safety across semantic, plan, and trajectory levels.
> >>
> >>**Role of semantic evaluation for LLM agents:** Given that the ground-truth safety constraints are necessarily defined in TL (as established above), our semantic-level evaluation is a process designed to measure an LLM agent's capability to correctly interpret safety rules (details of this process are explained at the beginning of **Section 2.3**). Empirically, we show that models with stronger TL grounding (e.g., Claude, DeepSeek) achieve higher plan- and trajectory-level safety, demonstrating the value of semantic evaluation in measuring an agent's safety literacy. Details of these experiments are provided in **Section 3.1** of the paper.

---

> ### Author Response · Authors · 2025-11-22
> **Response to Reviewer eNbM Part 2/4**
>
> > ## Doubts Regarding the Methodology:
> > #### Decoupled Hierarchy: The "multi-level" framework does not match an agent's runtime decision flow (Instruction$\rightarrow$Plan$\rightarrow$Action). "Level 1" (NL-to-TL translation) is merely an offline pre-processing step, completely decoupled from the agent's reasoning. A true hierarchical evaluation should first assess the agent's ability to identify risks from the initial instruction before planning.
> > #### Redundant Logic: The switch from LTL to CTL is poorly justified. Verifying that all n sampled trajectories are safe can be achieved by n independent LTL checks. Merging paths into a tree for a single, more complex CTL check is logically equivalent, and its added benefit over the simpler approach is not explained.
> > #### Offline-Only Assessment: The framework is a purely post-hoc auditing tool, not a real-time safety mechanism. It evaluates trajectories after they have been fully sampled, offering no capability to intervene or prevent an unsafe action as it happens, which severely limits its practical utility.
> >
> >> SENTINEL's multi-level verification pipeline is designed for safety evaluation, not as a replica of the agent's runtime decision flow. As stated in our response above, the semantic-level evaluation assesses whether an LLM agent can correctly interpret safety requirements, a metric that correlates with the downstream safety rate as shown in our experiments. There are also similar works in the literature that assess LLMs' capability in interpreting abstract logic-based semantics as a foundation for subsequent reasoning [1, 4-6].
> >>
> >> Regarding the usage of CTL, as stated in **Section 2.2**, checking a shared-prefix computation tree offers practical benefits over *n* independent LTL checks, including reduced redundancy, structured counterexample reporting, and compatibility with future branching-time and probabilistic extensions. In the revision, we have conducted additional experiments to compare our CTL checker's runtime against an LTL baseline. We benchmark both methods on three randomly selected tasks sampled from our task families (Pick-and-Place, Cool-and-Place, Heat-and-Place). For each task, we generate 100 trajectories per task following the same protocol as in **Section 3.3** and measure the evaluation duration, which includes parsing, merging (for CTL), and property checking. The LTL baseline uses the identical parser but evaluates each trajectory separately. The average runtime results below show that CTL is significantly faster than LTL, and this advantage becomes even more pronounced as the number of trajectories increases.
> >>
> >>| # Trajectories | CTL                         | LTL                   |
> >>| --------: | --------------------------- | --------------------- |
> >>|        10 | **$\mathbf{0.23_{\pm 0.06}}$**  | $1.72_{\pm 0.47}$     |
> >>|        25 | **$\mathbf{0.34_{\pm 0.08}}$**  | $4.35_{\pm 1.31}$     |
> >>|        50 | **$\mathbf{0.50_{\pm 0.16}}$**  | $8.96_{\pm 2.81}$     |
> >>|        75 | **$\mathbf{0.63_{\pm 0.16}}$**  | $12.95_{\pm 3.86}$    |
> >>|       100 | **$\mathbf{0.82_{\pm 0.22}}$**  | $18.00_{\pm 5.42}$    |
> >>
> >> Finally, SENTINEL is designed as an offline evaluation framework: similar to prior embodied safety benchmarks [10-11, 16-17], it focuses on post-hoc assessment. While it is possible to extend SENTINEL for real-time monitoring and shielding by leveraging the safety violation counterexamples it identifies, this is beyond the scope of this paper. We have also expanded the discussion in  **Section 4** on potential in-context learning and fine-tuning techniques for future safety refinement.

---

> ### Author Response · Authors · 2025-11-22
> **Response to Reviewer eNbM Part 3/4**
>
> > ## Lack of Evaluation on Mainstream VLA Models
> > #### Outdated Model Scope: The evaluation is restricted to LLMs, ignoring the more relevant Vision-Language-Action (VLA) models that dominate modern robotics. VLA models, which fuse vision into their decisions and directly output control params, will have different failure modes than text-only llms. By excluding VLAs, the paper fails to assess the framework's applicability to state-of-the-art agents or to compare safety performance across architectures.
> >>
> >> We acknowledge that VLA models represent an important class of embodied systems. However, our focus on LLM-based agents, as clearly defined in the **Abstract** and **Section 1 (Introduction)**, is a deliberate scope choice driven by the current research landscape and the specific strengths of our multi-level evaluation framework.
> >>
> >>First, a large body of recent embodied AI research, including VirtualHome [10], ALFWorld [11], Behavior-1K [2], EAI [1] and many LLM-as-planner frameworks, continues to evaluate text-driven agents [7-9, 12]. This is because LLMs provide interpretable, flexible, and modular high-level reasoning that can be adapted into diverse systems independent of specific policy architectures. In contrast, current VLA models tend to be end-to-end, task-specialized, and difficult to instrument with explicit safety specifications, making them less suitable for evaluating multi-level safety evaluations [12-15].
> >>
> >>Second, SENTINEL evaluates planning and decision-making safety, which is orthogonal to perception; the formal safety semantics we define (LTL/CTL) are model-agnostic and can be applied to any agent, including VLA models,  provided their plans and action traces can be extracted. One key challenge, however, lies in handling VLA models' continous control outputs. As noted in **Section 4** of the original manuscript, SENTINEL could be extended to support continuous control with formal safety semantics defined in languages such as Signal Temporal Logic (STL). However, efficient STL monitoring and continuous-signal verification [11,12] remain challenging for high-dimensional systems, and current explorations of foundation models with STL constraints [9,10] are still in the early stages. Therefore, extending SENTINEL to address continuous control with languages such as STL introduces significant technical challenges that are beyond the scope of this work and are left as future research.
>
>
> > ## Absence of Real-World Robotic Evaluation
> > #### Simulation-Only Validation: The experiments are confined to simulation, which is insufficient for evaluating physical safety. This approach ignores real-world physics, sensor noise, and actuator latencies that are critical to safety.
> > #### Ignoring the Sim-to-Real Gap: The paper makes no attempt to bridge the "sim-to-real gap." A plan verified as "safe" in simulation could fail catastrophically in the real world, and without validation on physical hardware, the claims about "physical safety" are not credible.
> >>
> >> Thank you for the comment. SENTINEL is the first framework designed to formally evaluate the safety of LLM-based embodied agents across semantic, plan, and trajectory levels. The primary contribution of this paper is establishing the foundation of this novel framework, by defining the formal safety semantics, developing the multi-level evaluation pipeline, and conducting experiments on LLM agents under various simulation scenarios. Given the increasing reliance on simulation for embodied AI development in practice, we believe that rigorous safety evaluation within these simulation environments is a critical and necessary step towards physical safety, and SENTINEL addresses exactly this need. We do not claim that the current SENTINEL alone would be sufficient to guarantee physical safety in real-world deployment, and we acknowledge the importance of addressing sim-to-real gap and conducting hardware validation. However, these are significant challenges in their own right and are beyond the scope of this paper. They are among the several promising extensions that can be built upon the current version of SENTINEL, as discussed in  **Section 4**.
> >>
> >>Moreover, the safety violations SENTINEL detects at the semantic and plan levels are fundamental understanding and reasoning failures that occur independent of most physical details. For example, if an LLM agent generates an unsafe action plan like heating metal in a microwave, this represents a critical reasoning failure that must be detected and addressed, regardless of finer-grained physical dynamics. In this regard, the current SENTINEL framework can serve as the ``first line of defense'' to filter out these logically-flawed agent designs before they ever reach physical hardware.

---

> ### Author Response · Authors · 2025-11-22
> **Response to Reviewer eNbM Part 4/4**
>
> > ## Missing Critical Experimental Details
> > #### Undefined Data Curation: The paper fails to describe the criteria or methodology used to extract the "safety-related subset" of tasks from VirtualHome and ALFRED. This makes it impossible to judge if the dataset is representative or biased.
> > #### Unspecified Dataset Size: The total size (N=?) of the task datasets is never reported. Without this, the reported percentage-based metrics are statistically meaningless and their robustness cannot be assessed.
> > #### Ambiguous Notation: Key formal notation is left ambiguous or undefined. For instance, the definition of "C" is not clearly defined in line 140-141, creating confusion in a paper that should be formally precise.
> >>
> >
> >> Thank you for these comments. We would like to clarify that task curation for VirtualHome and ALFRED is not performed manually; instead, tasks are extracted using explicit, simulator-derived criteria that align with the hazard categories defined in our taxonomy (heating, liquid–electronics, sharp objects, collision). These criteria and examples are detailed in **Appendix C.1 and Appendix C.2**. In the revision, we have updated the data size and property coverage statistics in **Appendix C.3**, and fixed the ambiguous notation issues.
> >>
> >>Finally, we want to emphasize that the primary focus of this work is on developing the formal safety evaluation pipeline across semantic, planning, and trajectory levels for LLM-based embodied agents, rather than building a large benchmark of safety constraints for various embodied tasks. We believe the latter can be achieved by domain experts leveraging the
> template-instantiation pipeline defined in **Appendix C.2**.
>
> > ## Reference
> >> [1]Li, Manling, et al. "Embodied agent interface: Benchmarking llms for embodied decision making." Advances in Neural Information Processing Systems 37 (2024): 100428-100534.\
> [2]Li, Chengshu, et al. "Behavior-1k: A benchmark for embodied ai with 1,000 everyday activities and realistic simulation." Conference on Robot Learning. PMLR, 2023.\
> [3]Zhou, Kaiwen, et al. "Multimodal situational safety." arXiv preprint arXiv:2410.06172 (2024).\
> [4]Han, Simeng, et al. "Folio: Natural language reasoning with first-order logic." Proceedings of the 2024 Conference on Empirical Methods in Natural Language Processing. 2024.\
> [5]Cheng, Fengxiang, et al. "Empowering llms with logical reasoning: A comprehensive survey." arXiv preprint arXiv:2502.15652 (2025).\
> [6]Toroghi, Armin, et al. "Verifiable, debuggable, and repairable commonsense logical reasoning via llm-based theory resolution." Proceedings of the 2024 Conference on Empirical Methods in Natural Language Processing. 2024.\
> [7]Chen, Yongchao, et al. "Autotamp: Autoregressive task and motion planning with llms as translators and checkers." 2024 IEEE International conference on robotics and automation (ICRA). IEEE, 2024.\
> [8]Chen, Yongchao, et al. "CodeSteer: Symbolic-Augmented Language Models via Code/Text Guidance." Forty-second International Conference on Machine Learning.\
> [9]Hao, Yilun, et al. "Large language models can solve real-world planning rigorously with formal verification tools." Proceedings of the 2025 Conference of the Nations of the Americas Chapter of the Association for Computational Linguistics: Human Language Technologies (Volume 1: Long Papers). 2025.\
> [10]Puig, Xavier, et al. "Virtualhome: Simulating household activities via programs." Proceedings of the IEEE conference on computer vision and pattern recognition. 2018.\
> [11]Shridhar, Mohit, et al. "Alfworld: Aligning text and embodied environments for interactive learning." arXiv preprint arXiv:2010.03768 (2020).\
> [12]Wu, Yi, et al. "SELP: Generating safe and efficient task plans for robot agents with large language models." 2025 IEEE International Conference on Robotics and Automation (ICRA). IEEE, 2025.\
> [13]Intelligence, Physical, et al. "π0. 5: a vision-language-action model with open-world generalization, 2025." URL https://arxiv. org/abs/2504.16054 1.2: 3.\
> [14]Zitkovich, Brianna, et al. "Rt-2: Vision-language-action models transfer web knowledge to robotic control." Conference on Robot Learning. PMLR, 2023.\
> [15]Zhang, Borong, et al. "SafeVLA: Towards Safety Alignment of Vision-Language-Action Model via Constrained Learning." arXiv preprint arXiv:2503.03480 (2025).\
> [16]Zihao Zhu, et al. “Earbench: Towards evaluating physical risk awareness for task planning of foundation model-based embodied
> ai agents.” arXiv preprint arXiv:2408.04449, 2024.\
> [17]Xiaoya Lu, et al. “Is-bench: Evaluating interactive safety of vlm-driven embodied agents in daily household tasks.” arXiv preprint arXiv:2506.16402, 2025.

---

> > ### Comment · Reviewer_eNbM · 2025-11-25
> >
> > Thank you for your detailed response. While I appreciate the clarifications on the methodology, specifically the computational efficiency of CTL, and the inclusion of missing experimental details, my fundamental concern regarding the limited scope and sim2real remains unresolved. The exclusion of VLA models and the lack of real-world validation significantly limit the framework's applicability to the current state-of-the-art in embodied AI. Given this remaining issue, I will maintain my original score.

---

> ### Author Response · Authors · 2025-11-26
>
> We appreciate the reviewer's engagement and are glad to note that our responses have addressed the comments regarding methodology and computational efficiency. However, we believe the remaining critiques regarding VLA coverage and sim-to-real expectations reflect a mismatch between the intended scope of the work and expectations for a robotics deployment paper, and we would like to reiterate the following points:
>
> **Regarding VLA models:**
> It is incorrect to claim that LLM-based agents are “outdated” or less relevant for the current embodied AI context. Many of the most recent and influential embodied-AI benchmarks [1-3,10,11,16,17] and planning methods [6-9,12] are LLM-based. Recent work [18] shows that LLM-based symbolic reasoning remains crucial for solving long-horizon, multi-step tasks, whereas current VLA models tend to specialize in short-horizon navigation or manipulation [19] and struggle with higher-level reasoning. Our choice to focus on LLM-based agents is therefore aligned with current practice, which has been clearly stated in the paper.
>
> **On real-world validation and sim-to-real:**
> This paper does not claim to fully resolve real-world physical safety; rather, it contributes a formal, simulation-based safety evaluation methodology, consistent with many widely-adopted pipelines in embodied AI, robotics, and reinforcement learning. Requiring real-robot deployment or sim-to-real adaptation in a framework-oriented paper that leverages existing simulators is not aligned with field norms. We emphasize that for specific robotic platforms, hardware integration and sim-to-real transfer would constitute substantial standalone efforts—orthogonal to the general-purpose evaluation design introduced here. Moreover, simulation-level safety evaluation is both meaningful and standard practice in robotics, RL, and control communities [20–22], as hardware-based safety testing is expensive, high-risk, and often infeasible during early-stage research, as had been elaborated in **Appendix B (Related Work)** of our original manuscript. SENTINEL provides the formal and modular foundation upon which subsequent systems, including real-world implementations, can be built.
>
> [18]Guruprasad, Pranav, et al. "Benchmarking vision, language, & action models on robotic learning tasks." arXiv preprint arXiv:2411.05821 (2024).\
> [19]Amin, Ali, et al. "$\pi^{*} _ {0.6} $: a VLA That Learns From Experience." arXiv preprint arXiv:2511.14759 (2025).\
> [20]Ji, Jiaming, et al. "Safety gymnasium: A unified safe reinforcement learning benchmark." Advances in Neural Information Processing Systems 36 (2023): 18964-18993.\
> [21]Dosovitskiy, Alexey, et al. "CARLA: An open urban driving simulator." Conference on robot learning. PMLR, 2017.\
> [22]Stevens, Brian L., Frank L. Lewis, and Eric N. Johnson. Aircraft control and simulation: dynamics, controls design, and autonomous systems. John Wiley & Sons, 2015.

---

### Official Review · Reviewer_Yq4o · 2025-10-30

**Soundness:** 2
**Presentation:** 2
**Contribution:** 2
**Rating:** 6
**Confidence:** 4

**Summary:**

SENTINEL translates natural language safety rules into temporal logic and evaluates LLM agents at three levels: semantic, plan, and trajectory. Multiple simulated rollouts are merged into a computation tree and checked with branching time operators so violations surface with concrete counterexamples. The framework is demonstrated in VirtualHome and ALFRED and the authors report a clear trade off where adding safety guidance improves safety but can reduce raw task success.

**Strengths:**

- Formal and end to end structure: The paper grounds safety in temporal logic and walks it through semantic interpretation, plan checks, and trajectory verification so you can see where failures originate and why.
- Branching time evaluation with actionable feedback: By assembling a computation tree from many rollouts and checking CTL operators, the method reasons about multiple possible outcomes and returns counterexample paths to diagnose hazards.
- Experiments across two simulators show that structured safety prompts improve safety and reveal a safety versus success trade off that can guide tuning.

**Weaknesses:**

- The paper performs CTL-style checking on a computation tree assembled from a finite set of sampled rollouts, primarily using universal CTL operators (such as $A\Phi$ or $E\Phi$). This is effectively *LTL on each branch with a universal/existential aggregation operation across the sampled tree*, so it neither reasons about unseen traces nor constitutes full CTL model checking of the underlying system (i.e. to my knowledge, it omits any nested path-quantified properties).
- The final trajectory level safety evaluation shows that even when the safety prompts given in LTL, the overall safety and success rates are in the single digit percentages. While it is shown that temporal logic given as part of the prompting process helps, it is not entirely convincing that this is enough for meaningful performance.
- Important extensions are future work. The paper notes that timed and probabilistic guarantees and integration with established checkers would broaden realism and rigor but are not part of the current system.
- Minor Presentation Concerns:
    - Temporal logic and LTL used interchangeably in the text.
    - L140: $C$ and $\mathcal{T}$ mixed?
    - L285: typo “sequences in top left” → in the
    - L287: typo “trajectories tree” → “tree of trajectories” / “trajectory tree”

**Questions:**

1. Can this  approach and its effectiveness be compared to SELP [1] ? They solve a similar task setting given an environment and task description in natural language, an LTL specification is extracted with tools like equivalence voting which is used to generate a plan better satisfying the given constraints.
2. In the LTL-based plan-level evaluation, is tracking whether an object undergoes a state change when moving from $s_0$ to $g$ mean that we assume full observability of all objects at all times or is this inferred by an LLM for unseen objects say out of range of the robot? If the later, how are misclassifications accounted for?
3. How is the accuracy of the generated LTL predicates verified given that it is extracted from the natural language task? How often do hallucinations cause problems?

### References:

[1] SELP: Generating Safe and Efficient Task Plans for Robot Agents with Large Language Models, Wu et al., ICRA 2025

---

> ### Author Response · Authors · 2025-11-22
> **Response to Reviewer Yq4o Part 1/4**
>
> Thank you for your time and constructive comments. We appreciate your recognition of our formal, end-to-end framework design for identifying safety failures and diagnosing their origins, our application of temporal logic and CTL-based checking, and our experimental findings. We provide our responses below to your comments and questions, and we have revised the manuscript with changes highlighted in blue text.
>
> > ## Conmment 1: The paper performs CTL-style checking on a computation tree assembled from a finite set of sampled rollouts, primarily using universal CTL operators (such as $A$ or $E$). This is effectively LTL on each branch with a universal/existential aggregation operation across the sampled tree, so it neither reasons about unseen traces nor constitutes full CTL model checking of the underlying system (i.e. to my knowledge, it omits any nested path-quantified properties).
> >>
> >> SENTINEL indeed performs CTL checking on a finite explicit computation tree constructed from sampled rollouts, and therefore evaluates CTL formulas over this tree, rather than over the full underlying embodied system. As discussed in Remark 2.1, applying full-state CTL model checking on a realistic embodied household environment would lead to severe state space explosion and require substantial computational overhead in converting embodied dynamics into symbolic encoding such as Binary Decision Diagrams. On the other hand, there are effective approaches within control and formal methods domain to conduct checking on trajectories [6,7] for safety and reachability analysis. Considering these, the goal of this work is to provide sound checking with respect to the sampled tree, evaluating whether all observed executions satisfy the specified properties, rather than to exhaustively verify the entire simulator dynamics symbolically.
> >>
> >> To verify the finite sample trajectories, CTL-based checking has advantages over LTL in terms of efficiency. We have conducted an additional experiment to compare our CTL checker's runtime against an LTL baseline (please see **Section 3.4** in the revision for details). In particular, our CTL checker merges all trajectories for a given task into a singular tree and then performs a tree search, whereas the LTL baseline checks each trajectory independently in sequence using the same parser. Our design of the CTL checker leverages the advantage of computation tree over pure LTL-based checking in exploiting redundant states and traces and in recursive searches [2]. We benchmark both checkers on three randomly selected tasks sampled from our task families (Pick-and-Place, Cool-and-Place, Heat-and-Place). For each task, we generate 100 trajectories per task following the same protocol as in **Section 3.3** and measure the evaluation duration which includes parsing, merging (for CTL), and property checking. The LTL baseline uses the identical parser but evaluates each trajectory separately. The average runtime results below show that CTL is significantly faster than LTL, and this advantage becomes even more pronounced as the number of trajectories increases.
> >>
> >>| # Trajectories | CTL                         | LTL                   |
> >>| --------: | --------------------------- | --------------------- |
> >>|        10 | **$\mathbf{0.23_{\pm 0.06}}$**  | $1.72_{\pm 0.47}$     |
> >>|        25 | **$\mathbf{0.34_{\pm 0.08}}$**  | $4.35_{\pm 1.31}$     |
> >>|        50 | **$\mathbf{0.50_{\pm 0.16}}$**  | $8.96_{\pm 2.81}$     |
> >>|        75 | **$\mathbf{0.63_{\pm 0.16}}$**  | $12.95_{\pm 3.86}$    |
> >>|       100 | **$\mathbf{0.82_{\pm 0.22}}$**  | $18.00_{\pm 5.42}$    |
> >>
> >> Finally, regarding nested path-quantified properties: while our CTL parsing and operator implementation support standard path quantification and checking of nested path quantifiers, those properties do not offer additional expressive benefit on finite rollout trees and are therefore not included in our defined CTL subset for safety constraints. We have clarified this in the revision in **Section 2.2**.

---

> ### Author Response · Authors · 2025-11-22
> **Response to Reviewer Yq4o Part 2/4**
>
> > ## Comment 2: The final trajectory level safety evaluation shows that even when the safety prompts given in LTL, the overall safety and success rates are in the single digit percentages. While it is shown that temporal logic given as part of the prompting process helps, it is not entirely convincing that this is enough for meaningful performance.
> >
> >> We appreciate this comment regarding the results, which we also found interesting and discussed in the original manuscript (**Section 3.3** and **Appendix F.2**). However, we would first like to emphasize that these results reflect the limitations of current LLM agents (for high-level action planning) and standard safety-agnostic low-level controllers, rather than a limitation of SENTINEL. The goal of SENTINEL currently focuses on **evaluating** the safety of given LLM-based embodied agents across semantic, planning, and trajectory levels, rather than trying to improve their safety. In addition, SENTINEL is designed to be flexible such that users can introduce their own safety constraint formulations and prompting contents for evaluating those LLM-based embodied agents with formal semantics.
> >>
> >> The low trajectory-level safety rate suggests that, even when we supply explicit temporal-logic safety guidance in the prompt, current LLM agents and standard safety-agnostic low-level controllers used in simulation environments still frequently fail to comply with physical safety constraints. As we discussed in **Section 3.3** and **Appendix F.2**, these failures arise from a mixture of causes: the LLM agents may generate unsafe high-level action orders; and, more commonly, the low-level controllers used in simulators lack built-in safety awareness and often generate unsafe trajectories.
> >>
> >> In the original manuscript, we included a case study in **Appendix F.2**, where the LLM agent produces a semantically-reasonable and safety-aware plan, yet the final trajectory is unsafe due to how low-level trajectories are generated without safety consideration. In the revision, to further investigate this aspect, we extended the analysis in **Appendix F.2**
> by introducing a simple heuristic low-level \emph{safety controller} and evaluating its effectiveness. We focus on a subset of \emph{Pick-and-Place} tasks involving placing a candle at a target location. Instead of directly mapping the LLM agent's high-level action sequence plan to low-level controls, we insert a safety-aware shield: when the agent is holding a candle, if a flammable object is seen nearby, the low-level safety controller immediately executes an action to extinguish the candle before proceeding with the next LLM-agent-specified action. This mimics a real-world setup where a robot arm has an end-effector sensor for imminent collision or hazard detection.
> We evaluated this modified low-level controller across all 20 candle-related tasks. While the simple heuristic improves the safety in some cases, it still fails in the majority of the scenarios.
> The agent still struggles to detect fire hazards when flammable objects are not directly visible (e.g., due to rotation or camera horizon), and the shield does not address other active safety constraints such as open-door collisions or hand-object collisions. Moreover, while the heuristic improves safety in some scenarios, it is not a one-size-fits-all solution even within the candle-related tasks. For instance, when the task requires bringing a lit candle to a table that already contains flammable objects for illumination, a more appropriate strategy would be to first remove the flammable items rather than forcefully extinguishing the candle. This example further demonstrates the challenges in meeting safety constraints across plan and trajectory levels, showing **the importance of having a systematic evaluation framework like SENTINEL**.

---

> ### Author Response · Authors · 2025-11-22
> **Response to Reviewer Yq4o Part 3/4**
>
> > ## Comment 3: Important extensions are future work. The paper notes that timed and probabilistic guarantees and integration with established checkers would broaden realism and rigor but are not part of the current system.
> >>
> >> We agree that these directions are valuable and significant. However, the primary contribution of this work is to establish the **first framework** for formally evaluating the physical safety of LLM-based embodied agents, by defining the formal safety semantics, developing the foundational evaluation pipeline across semantic, plan, and trajectory levels, and conducting experiments on a variety of LLM agents under various scenarios. The current version of SENTINEL serves as the foundation upon which various future extensions, such as those discussed in **Section 4** (including multi-agent systems, timed safety properties, continuous behavior, probabilistic settings, etc.), can be built. Note that even without these extensions, the current framework is already applicable to a variety of systems and safety properties and effective in identifying critical safety violations. For example, if an LLM agent generates unsafe action plans like placing metal in a microwave, this represents a critical logic failure that must be detected and addressed, regardless of the finer-grained continuous dynamics or timing details.
> >>
> >> Moreover, extensions such as addressing timed safety properties and probabilistic settings represent significant research undertakings in their own right. They will require richer temporal models, continuous-time state representations, specialized monitoring algorithms, and new encoding and verification techniques to address the associated complexities that far exceed the scope of this paper.
>
> > ## Comment 4: Minor Presentation Concerns
> >>
> >> Thank you for pointing out these presentation issues. We have fixed them in the revision.
>
> > ## Question 1: Can this approach and its effectiveness be compared to SELP [1] ? They solve a similar task setting given an environment and task description in natural language, an LTL specification is extracted with tools like equivalence voting which is used to generate a plan better satisfying the given constraints.
> >>
> >> We appreciate the reviewer for mentioning this relevant work. Unlike SELP, which conducts LLM-based task planning with safety improvement, our framework focuses on the safety evaluation of given LLM-based embodied agents. Works like SELP are complementary to our approach and could potentially be combined with our framework for future safety refinement. We have expanded **Section 4** in the revision to discuss this direction, and included SELP in the references.
>
> > ## Question 2: In the LTL-based plan-level evaluation, is tracking whether an object undergoes a state change when moving from $s_0$ to $g$ mean that we assume full observability of all objects at all times or is this inferred by an LLM for unseen objects say out of range of the robot? If the later, how are misclassifications accounted for?
> >>
> >> We assume full observability for plan-level LTL evaluation where object states are obtained from the simulator's metadata. Our goal at this level is to verify the logical consistency of the agent's high-level plan rather than its perception or inference abilities.
>
> > ## Question 3: How is the accuracy of the generated LTL predicates verified given that it is extracted from the natural language task? How often do hallucinations cause problems?
> >>
> >> The LTL predicates used for safety checking are not generated by LLMs; rather, the atomic propositions/predicates are manually defined or provided from the metadata of simulators. Similar practices are used in the literature [3-5]. Since the set of atomic propositions is provided at the semantic evaluation level, hallucinations do not occur in terms of wrong propositions, which has also been investigated by [4].

---

> ### Author Response · Authors · 2025-11-22
> **Response to Reviewer Yq4o Part 4/4**
>
> > ## Reference
> >> [1]Wu, Yi, et al. "SELP: Generating safe and efficient task plans for robot agents with large language models." 2025 IEEE International Conference on Robotics and Automation (ICRA). IEEE, 2025.\
> [2]Baier, Christel, et al. "Principles of model checking." MIT press, 2008.\
> [3]Chen, Yongchao, et al. "Nl2tl: Transforming natural languages to temporal logics
> using large language models." arXiv preprint arXiv:2305.07766 (2023).\
> [4]Liu, Jason Xinyu, et al. "Lang2ltl: Translating natural language commands to temporal specification with large language models." Workshop on Language and Robotics
> at CoRL 2022. 2022.\
> [5]Li, Manling, et al. "Embodied agent interface: Benchmarking llms for embodied decision making." Advances in Neural Information Processing Systems 37 (2024):
> 100428-100534.\
> [6]Desai, Ankush, et al. "Combining model checking and runtime verification for safe robotics." International Conference on Runtime Verification. Cham: Springer International Publishing, 2017.\
> [7]Giorgetti, Nicolo, et al. "Bounded model checking of hybrid dynamical systems." Proceedings of the 44th IEEE Conference on Decision and Control. IEEE, 2005.

---

### Official Review · Reviewer_gPme · 2025-11-02

**Soundness:** 3
**Presentation:** 2
**Contribution:** 4
**Rating:** 6
**Confidence:** 3

**Summary:**

This paper presents a novel evaluation framework for assessing the safety of LLM-based embodied agents, with three key contributions:

1. **Formal Safety Specification**  It proposes grounding the safety specification in formal temporal logics, specifically Linear Temporal Logic (LTL) and Computation Tree Logic (CTL).

2. **Multi-Level Safety Evaluation Framework**  The framework evaluates safety from three distinct perspectives:

   - **Semantic Level**
     Assesses the ability of LLM agents to understand natural language safety constraints by checking whether their generated temporal logic expressions are semantically equivalent to ground truth expressions.

   - **Plan Level**
     Formally verifies that the high-level plans generated by LLMs do not violate any of the formal safety constraints.

   - **Execution Trace Level**
     Evaluates whether the detailed execution trajectories of the agents satisfy all safety constraints.

3. **Experimental Evaluation**
   A detailed experimental study highlights the strengths and weaknesses of state-of-the-art LLMs in adhering to safety constraints.

**Strengths:**

1. **Relevance**
The paper addresses a critical challenge: assessing the safety of AI systems (in particular they ability to understand and follow safety constraints).

2. **Significance and Novelty**
To the best of my knowledge, the proposed approach is novel in its problem formulation and its reliance on formal temporal logics to formally capture and check safety constraints. The introduction of this framework holds strong potential for significant impact, as it provides a vital resource that could accelerate progress in the safe LLM-based embodied agents

3. **Experimental Evaluation**
The experimental results clearly highlight the strengths and the limitations of existing state-of-the-art LLMs in their ability to understand and successfully adhere safety constraints. In particular,  the safety assessment at the level of detailed execution trajectories shows that even the best frontier models mostly generate unsafe execution trajectories ( % of successful and safe trajectories is less than 5).

**Weaknesses:**

1. [Poor Presentation] Although I believe that the proposed approach is sound,  the presentation of the main technical sections related to problem statement and temporal logics (sections 2.1 and 2.2) can be significantly improved. A lot of important concepts introduced, but they are either not formally defined at all or defined much later in the paper. Here are a few examples:
 - the authors write:
 > These trajectories are **merged into a computation tree T** , and CTL-based checking is applied to verify that **C** holds across all possible execution branches ...
  **C* is never defined. What is **C**? The mechanism to **merged  [trajectories] into a computation tree T** is only explained 3 page later (in page 6).
  - Another example, in line 157, the authors use the term "labeling function L" for the first time without defining it (again the formal definition is provided only 3 page later).
  -  Finally in line 158, the authors write "σ |= a iff p ∈ L(s_0))". What is a? What is the relationship between a and p?

2. Some minor issues:
 - In line 170, "AGφ (a safety invariant: φ holds on all paths)" should be replaced with AGφ (a safety invariant: φ **always** holds on all paths)
 - In line 398-399, "Overall results are reported in **Section 3.2**." should be replaced with "Overall results are reported in **Table 3**."
 - In line 457, "Results are summarized in **Section 3.3**." should be replaced with ""Results are summarized in **Table 4**."

**Questions:**

I have asked all my questions in the weaknesses section

---

> ### Author Response · Authors · 2025-11-21
> **Response to Reviewer gPme**
>
> Thank you for your time and constructive comments. We appreciate your recognition of our work's relevance, novelty, and significance, especially its potential impact on building safe LLM-based embodied agents. We have revised the manuscript accordingly, with changes highlighted in blue.
>
> > ## Comment 1: [Poor Presentation] Although I believe that the proposed approach is sound, the presentation of the main technical sections related to problem statement and temporal logics (sections 2.1 and 2.2) can be significantly improved. A lot of important concepts introduced, but they are either not formally defined at all or defined much later in the paper. Here are a few examples:
> >>
> >> We thank the reviewer for the suggestions on the presentation section. We have made the following changes:
> >> - Moving the formal definitions of trajectory tree $\mathcal{T}$ and labeling function $L$ to **Section 2.2**
> >> - Fixing typo of $a$ in line 157 to $p$, which should stands for atomic proposition
> > ## Comment 2: Some minor issues (Writing and Notations).
> >>
> >> Thank you for the detailed comments. We have fixed all of them in the revision.

---

### Official Review · Reviewer_g5BS · 2025-11-03

**Soundness:** 2
**Presentation:** 3
**Contribution:** 3
**Rating:** 4
**Confidence:** 4

**Summary:**

This paper introduces SENTINEL, a multi-level safety evaluation framework for LLM-based embodied agents using temporal logics. Safety requirements are encoded in LTL/CTL and checked at three levels: semantic (NL to LTL), plan-level (LTL over high-level plans), and trajectory (CTL over computation trees). The framework is evaluated on VirtualHome and ALFRED, and across several open and closed LLM families.

**Strengths:**

(+) The paper provides a framework to encode safety specifications into temporal logic semantics LTL/CTL, which allow more correct representation and support formal verification afterwards. This is promising for guaranteeing safety of LLMs and interpretability.
(+) Verification is conducted at multiple levels including semantic, plan, and trajectory, providing stricter safety guarantees than related works and preventing unsafe actions during agent interactions.
(+) The paper is well-organized and easy to follow.

**Weaknesses:**

(-) Though the idea of using formal logic checking is interesting, there are several weaknesses in the current implementation. While SENTINEL reimplements CTL checking (via BFS/DFS traversal), it does not leverage existing mature model-checking tools (e.g., NuSMV, SPIN, PRISM). Although the paper's key difference lies in its use of formal verification, it has not demonstrated that SENTINEL's in-house LTL/CTL checker is equivalent to standard model-checking semantics. It defines satisfaction rules for LTL/CTL from natural languages but it's still unclear to me that the verification is sound (no false positives) or complete (no missed violations). To me, the claimed rigor of "formal verification" remains conceptual rather than guaranteed.

(-) No refinement process is used, i.e., the framework performs plan-level verification in isolation without progressively refining or strengthening specifications during execution. Consequently, plan-level checks cannot capture runtime or spatially dependent conditions (e.g., proximity, force, timing).

(-) Evaluation is confined to VirtualHome and ALFRED, both using discrete action spaces. The framework is not tested in continuous, more complicated environments where continuous control and real-time safety verification are critical, limiting generalizability to real-world application.

(-) The framework depends on a curated set of ground-truth temporal-logic constraints per task, but the paper only briefly describes how these constraints were created. It does not explain who curated them, how long it took, or how correctness and consistency were validated. This makes the specification process hard to reproduce or adapt to new domains that require expert-defined safety rules.

(-) The paper lacks quantitative analysis of property coverage and verification efficiency. It does not report how many properties were tested per task, their representativeness across safety categories, or the average number verified. Verification time, storage overhead, and scalability as trajectory complexity grows are also not discussed, leaving the practical scalability and completeness of the method unclear.

Suggestions:
1. Table (II) should include a column describing the capability of the compared models (i.e., performance under MMLU benchmark) for better comparison.
2. Providing suggestions on how to leverage verification results/feedback to help agents improve their safety/satisfaction rate (e.g., iterative repair, refinement, or safety-aware planning) would further enhance the practical value of the framework and be helpful to the community.

**Questions:**

Please clarify the negative points above, specifically around soundness of analysis results.

---

> ### Author Response · Authors · 2025-11-21
> **Response to Reviewer g5BS Part 1/4**
>
> > ## Comment 1: Though the idea of using formal logic checking is interesting, there are several weaknesses in the current implementation. While SENTINEL reimplements CTL checking (via BFS/DFS traversal), it does not leverage existing mature model-checking tools (e.g., NuSMV, SPIN, PRISM). Although the paper's key difference lies in its use of formal verification, it has not demonstrated that SENTINEL's in-house LTL/CTL checker is equivalent to standard model-checking semantics. It defines satisfaction rules for LTL/CTL from natural languages but it's still unclear to me that the verification is sound (no false positives) or complete (no missed violations). To me, the claimed rigor of "formal verification" remains conceptual rather than guaranteed.
> >>
> >> We appreciate the comment regarding the use of mature model-checking tools and the question on our checker's soundness and completeness. Below, we clarify our design choices and the theoretical grounding of our framework.
> >>
> >> **Choice of developing a custom checker vs. using existing tools:** While mature tools such as PRISM [1], SPIN [2] and STORM [7] are powerful, integrating them into current embodied simulation environments would introduce substantial overhead. These tools typically require symbolic encoding of environment dynamics (e.g., Binary Decision Diagrams in NuSMV and PRISM) and atomic propositions as input, which become computationally prohibitive and prone to state-space explosion in detailed embodied settings [6]. For instance, NuSVM and SPIN can only handle systems with between 100 to 300 state variables [19], and PRISM is limited to navigations tasks for its robotic applications [20]. Therefore, we develop a lightweight, explicit-state checker for our current problem setting (as explained more below). For future work involving probabilistic settings, we do agree that adapting tools such as PRISM [1] with suitable abstraction could be beneficial, as noted in Remark 2.4 and Section 4 of the original manuscript. We have further expanded **Remark 2.4** in the revision to clarify our design choice.
> >>
> >>**Soundness and completeness of SENTINEL checker:** We first would like to clarify that SENTINEL does not introduce new or approximate semantics for LTL/CTL. Our checker implements a subset of the standard textbook model-checking semantics [6] directly over a finite trajectory tree from complex embodied systems, not the symbolic representation of the systems. This is common for model checking in robotic control context [17, 18]. Each operator is evaluated exactly based on classical recursive computation (essentially boils down to bottom-up traversal algorithms) [6], which is similar to algorithmic procedures used by tools like PRISM [1] and SPIN [2] when given explicit deterministic finite trajectories. As our checking algorithm follows the procedures outlined in the standard literature [6], the verification is **sound** with respect to the given trajectory tree and CTL specification (i.e., it yields no false positives). As for completeness, our CTL checker is designed to return the first encountered counterexample and terminate, similar to  existing model checkers listed above. It is **complete** with respect to falsification: if the trajectory tree contains any violations, the checker is guaranteed to detect one and report the system as unsafe. It does not report all the violations, although that could be implemented at the expense of higher computational complexity.
> >>
> >> Finally, we would like to highlight that the core contribution of this work is not the development of new formal verification techniques, but rather the novel application of formal methods to establish a multi-level safety evaluation framework for LLM-based embodied agents. The proposed evaluation framework is more systematic, comprehensive, and reproducible than existing methods in the literature (e.g., those listed in Table 1 of the paper).

---

> ### Author Response · Authors · 2025-11-21
> **Response to Reviewer g5BS Part 2/4**
>
> > ## Comment 2 & Suggestion 1: No refinement process is used, i.e., the framework performs plan-level verification in isolation without progressively refining or strengthening specifications during execution. Consequently, plan-level checks cannot capture runtime or spatially dependent conditions (e.g., proximity, force, timing).
> >>
> >> Thank you for the comment. We would like to first emphasize that the current scope of SENTINEL is to serve as a safety **evaluation** framework, rather than an online shielding or iterative planning refinement mechanism. In our evaluation design, plan-level checking intentionally focuses on symbolic temporal dependencies, while physics- and runtime-dependent hazards (e.g., proximity, force) are captured at the trajectory level through CTL verification over simulator-generated trajectories. This division enables direct evaluation of the agent's ability to interpret and address safety constraints during its reasoning process, before simulation inaccuracies or limitations of lower-level controllers complicate the picture. The motivation for this design is detailed in the original manuscript in **Section 2.3 (above Remark 2.4)**. Additionally, we provide case studies in **Appendix F.2** and the newly-added **Appendix F.3**  in the revision to illustrate why both plan-level and trajectory-level checks are necessary and complementary.
> >>
> >> We do agree that refinement would be a valuable application of our framework in future work. While SENTINEL currently functions as an evaluator for given agent designs, its diagnostic output can provide important signals to drive safety improvements. In the revision, we have expanded **Section 4** to discuss how SENTINEL can provide feedback to enable safety-centric refinement in future agent designs.
>
> > ## Comment 3: Evaluation is confined to VirtualHome and ALFRED, both using discrete action spaces. The framework is not tested in continuous, more complicated environments where continuous control and real-time safety verification are critical, limiting generalization to real-world application.
> >>
> >> Thank you for the constructive comment. While ALFRED indeed uses discrete high-level PDDL actions, its underlying engine, AI2-THOR, supports continuous physics simulations (e.g., forces, object dynamics, temperature, and liquid states). As noted in the original manuscript in **Section 4**, SENTINEL could be extended to support continuous control with formal safety semantics defined in languages such as the Signal Temporal Logic (STL), whose robustness metrics and real-valued predicates are well-suited for continuous trajectories. However, efficient STL monitoring and continuous-signal verification [11,12] remain challenging for high-dimensional systems, and current explorations of LLMs with STL constraints [9,10] are still in the early stages. Therefore, extending SENTINEL to address continuous control with languages such as STL introduces significant technical challenges that are beyond the scope of this work and are left as future research.
> >>
> >> Regarding real-time support, enabling real-time verification requires simulators to provide precise temporal progression and event scheduling. While simulators such as AI2-THOR and BEHAVIOR-1K are built on relatively realistic physics, they currently lack the full real-time scheduling and multi-object temporal monitoring capabilities needed for real-time safety evaluation. We discussed this limitation in the original manuscript (**Remark 2.1**), and consider real-time safety verification as a natural extension of SENTINEL for future work.

---

> ### Author Response · Authors · 2025-11-21
> **Response to Reviewer g5BS Part 3/4**
>
> > ## Comment 5: The paper lacks quantitative analysis of property coverage and verification efficiency. It does not report how many properties were tested per task, their representativeness across safety categories, or the average number verified. Verification time, storage overhead, and scalability as trajectory complexity grows are also not discussed, leaving the practical scalability and completeness of the method unclear.
> >>
> >> Thank you for the constructive comment. We have conducted two new experiments to evaluate the efficiency and scalability of our CTL checker, as detailed below.
> >>
> >>**Efficiency:** First, we compare the runtime of our CTL checker against an LTL checker. Our CTL checker merges all trajectories for a given task into a singular tree and then performs a tree search, whereas the baseline LTL checker examines each trajectory independently in a sequential order using the same parser. This design signifies the advantage of computation tree at exploiting redundant states and recursive searches compared to a pure LTL checker [6].
> >> Concretely, we benchmark both algorithms on three randomly-selected tasks from our task families (Pick-and-Place, Pick-Cool-and-Place, Pick-Heat-and-Place). We generate 100 trajectories per task following the same protocol as in **Section 3.3** and measure the evaluation duration which includes parsing, merging (for CTL), and property checking. The average runtimes (in seconds) are:
> >>
> >>| # Trajectories | CTL                         | LTL                   |
> >>| --------: | --------------------------- | --------------------- |
> >>|        10 | **$\mathbf{0.23_{\pm 0.06}}$**  | $1.72_{\pm 0.47}$  |
> >>|        25 | **$\mathbf{0.34_{\pm 0.08}}$**  | $4.35_{\pm 1.31}$     |
> >>|        50 | **$\mathbf{0.50_{\pm 0.16}}$**  | $8.96_{\pm 2.81}$     |
> >>|        75 | **$\mathbf{0.63_{\pm 0.16}}$**  | $12.95_{\pm 3.86}$    |
> >>|       100 | **$\mathbf{0.82_{\pm 0.22}}$**  | $18.00_{\pm 5.42}$    |
> >>
> >> As the table shows, our CTL checker is significantly faster than the LTL baseline for the same set of trajectories, and the gap grows with the number of trajectories. This assessment on verification time demonstrates that the CTL-based approach is practically more efficient.
> >>
> >>**Scalability:** Second, we evaluate scalability on real execution logs from ALFRED across all 91 modified tasks and 4 different LLMs. Using the stored trajectories from our main experiments, we re-run the CTL safety checker while varying the number of constraints from 10, 25, 50, 75, 100, to an extreme case of 500 constraints. For 10–100 constraints, we select subsets of the existing safety rules whereas for the 500-constraint setting, we augment the rule set with unique placeholder safety constraints so that the checker must still parse and evaluate the entire tree for all 500 formulas. We then measure the evaluation time for each setting. The aggregate runtime statistics are reported below. For 10–100 constraints, the mean evaluation time remains well under 0.5 seconds even for long-horizon tasks with over 70 executed steps in the simulator. Even in the extreme 500-constraint setting, the average runtime is only about **1.07 second**. These results show that verification time grows gently with the number of constraints and remains low in absolute terms for our CTL checker, demonstrating that it scales efficiently to large sets of safety specifications across all tasks and models considered.
> >>
> >>| # Constraints | CTL                     |
> >>| ------------: | ---------------------------- |
> >>|            10 | **$\mathbf{0.0518_{\pm 0.0342}}$** |
> >>|            25 | **$\mathbf{0.0879_{\pm 0.0535}}$** |
> >>|            50 | **$\mathbf{0.1413_{\pm 0.0811}}$** |
> >>|            75 | **$\mathbf{0.1909_{\pm 0.1063}}$** |
> >>|           100 | **$\mathbf{0.2425_{\pm 0.1347}}$** |
> >>|           500 | **$\mathbf{1.0693_{\pm 0.5880}}$** |
> >>
> >>**Property Coverage:** Additionally, we have extended the property coverage in the revision (please see **Appendix C.3** for details). Note that as stated above, the focus of this work is on developing the formal safety evaluation pipeline for LLM-based embodied agents rather than building a large benchmark of safety constraints.

---

> ### Author Response · Authors · 2025-11-21
> **Response to Reviewer g5BS Part 4/4**
>
> > ## Comment 4: The framework depends on a curated set of ground-truth temporal-logic constraints per task, but the paper only briefly describes how these constraints were created. It does not explain who curated them, how long it took, or how correctness and consistency were validated. This makes the specification process hard to reproduce or adapt to new domains that require expert-defined safety rules.
> >>
> >> Thank you for the comment. We developed a template-instantiation pipeline for generating the ground-truth temporal-logic constraints: first, (1) a small library of general LTL patterns is created to cover common hazard types (e.g., heating, liquid-electronics, sharp objects, collision) in the target embodied environments. Then, (2) temporal-logic constraints are automatically instantiated based on these patterns and simulator metadata such as object category, receptacle type, and task type. Note that in practice, the template library should be created by experts with knowledge in temporal logic and target embodied environments, following standard practices in prior work [13–16]. We have expanded **Appendix C.2** in the revision with details and examples to make this process clear and reproducible.
> >>
> >>We would also like to emphasize that the focus of this work is on developing the formal safety evaluation pipeline across semantic, planning, and trajectory levels for LLM-based embodied agents, rather than building a large benchmark of safety constraints for various embodied tasks. We believe the latter can be achieved with domain experts leveraging the template-instantiation pipeline listed in our framework.
>
> > ## Suggestions
> >>
> >> We appreciate the constructive suggestions. In the revision, we have added additional column in **Table 2** to show base models' foundational capability, and we have expanded the discussion in **Section 4** on potential directions to leverage verification results/feedback to help agents improve their safety/satisfaction rate.
>
> > ## Reference:
> >> [1]Marta Kwiatkowska, et al. "Prism: Probabilistic symbolic model checker." In International Conference on Modelling Techniques and Tools for Computer Performance Evaluation.\
> [2]Holzmann, Gerard J. “The SPIN model checker: Primer and reference manual.” Reading: Addison-Wesley, 2004.\
> [3]Mascle, Corto, et al. "From LTL to rLTL monitoring: improved monitorability through robust semantics." Proceedings of the 23rd International Conference on Hybrid Systems: Computation and Control.\
> [4]David, Alexandre, et al. "Uppaal stratego." International Conference on Tools and Algorithms for the Construction and Analysis of Systems.\
> [5]Li, Chengshu, et al. "Behavior-1k: A benchmark for embodied ai with 1,000 everyday activities and realistic simulation." Conference on Robot Learning, 2023.\
> [6]Baier, Christel, et al. "Principles of model checking." MIT press, 2008.\
> [7]Dehnert, Christian, et al. "A storm is coming: A modern probabilistic model checker." International Conference on Computer Aided Verification, 2017.\
> [8]Maler, Oded, et al. "Monitoring temporal properties of continuous signals." International symposium on formal techniques in real-time and fault-tolerant systems.\
> [9]He, Jie, et al. "Deepstl: from english requirements to signal temporal logic." Proceedings of the 44th International Conference on Software Engineering. \
> [10]Fang, Yue, et al. "Enhancing Transformation from Natural Language to Signal Temporal Logic Using LLMs with Diverse External Knowledge."\
> [11] Donzé, Alexandre, et al. "Efficient robust monitoring for STL." International conference on computer aided verification.\
> [12]Ničković, Dejan, et al. "RTAMT: Online robustness monitors from STL." International Symposium on Automated Technology for Verification and Analysis. Cham: Springer International Publishing, 2020.\
> [13]Fuggitti, Francesco, et al. "Nl2ltl–a python package for converting natural language instructions to linear temporal logic formulas." Proceedings of the AAAI Conference on Artificial Intelligence 2023.\
> [14]Chen, Yongchao, et al. "Nl2tl: Transforming natural languages to temporal logics using large language models."\
> [15]Liu, Jason Xinyu, et al. "Lang2ltl: Translating natural language commands to temporal specification with large language models." Workshop on Language and Robotics at CoRL 2022.\
> [16]Li, Manling, et al. "Embodied agent interface: Benchmarking llms for embodied decision making." Advances in Neural Information Processing Systems 37.\
> [17]Desai, Ankush, et al. "Combining model checking and runtime verification for safe robotics." International Conference on Runtime Verification.\
> [18]Goppert, James, et al. "Model checking of a flapping-wing mirco-air-vehicle trajectory tracking controller subject to disturbances." \
> [19]Clarke Jr, Edmund M. "Model Checking Overview."\
> [20]Lacerda, Bruno, et al. "Probabilistic planning with formal performance guarantees for mobile service robots." The International Journal of Robotics Research.

---

### Meta-Review · Area_Chair_hvJT · 2026-01-02

**Summary:**

This paper presents SENTINEL, a novel framework for formal safety evaluation of LLM-based embodied agents across semantic, plan, and trajectory levels. The reviewers generally acknowledge the technical completeness and innovation of the proposed approach, particularly its use of formal logic checking and the three-level verification pipeline. While some reviewers questioned the motivation for employing logical checking instead of natural language-based evaluation, the authors provided satisfactory clarification in their rebuttal regarding the advantages of formal semantics in ensuring precise safety specifications.

However, two major concerns remain significant. First, the current framework operates as an offline evaluation system, which cannot provide real-time safety guarantees during the actual operation of embodied agents. Second, the work focuses exclusively on LLM-based agents without considering Vision-Language Models (VLMs), thereby limiting safety assessment to the planning level and discrete action domains. This restriction creates a substantial gap between the proposed evaluation framework and practical physical safety assessment in real-world scenarios.

Although the authors attempted to justify the reasonableness of their setting and present the possible future extension in their response, these concerns represent fundamental shortcomings that are difficult to overlook. Given the existing body of work on planning-level evaluation, this paper does not substantially advance the field of embodied agent safety assessment. The contribution appears somewhat incremental rather than transformative.

We encourage the authors to consider these valuable suggestions in their future work, particularly by extending formal logic checking to enable genuine safety evaluation in physical environments, which would represent a more significant advancement in this research domain.

**Reviewer Concerns:**

please refer to the summary

**Reviewer Scores:**

please refer to the summary

---

### Decision · Program_Chairs · 2026-01-26

Reject